# Dilemmas Related to Young Children's Participation and Rights: A Discourse Analysis Study of Present and Future Professionals Working with Children

Eija Sevón *, Marleena Mustola and Maarit Alasuutari

Department of Education, University of Jyväskylä, 40014 Jyväskylä, Finland; marleena.mustola@jyu.fi (M.M.); maarit.alasuutari@jyu.fi (M.A.)
* Correspondence: eija.sevon@jyu.fi

**Abstract:** According to the United Nations Convention on the Rights of the Child (UNCRC), every child has the right to be heard and express their views in matters that concern them. Yet, participation is one of the most debated aspects of the UNCRC. Although children's participation is a statutory requirement of Finnish early childhood education and care (ECEC) and schools, educators are often unfamiliar with how to meet the demands of participation. In this study, we examined what kinds of counter discourses about the realization of children's participation could be differentiated in interviews with present and future education professionals who took part in a study program focusing on knowledge and skills regarding young children's rights and participation. The data, which consisted of individual and group interviews with 31 participants, were analyzed with discourse analysis. Three counter discourses were identified: unrealized, adult-defined, and elusive participation. The discourses illuminated various dilemmas in children's participation. Awareness of such dilemmas enables the development of pedagogical practices that enhance children's wellbeing and rights.

**Keywords:** children's citizenship; children's rights; discourse analysis; early childhood education and care; equity; Finland; participation





## 1. Introduction

Questions related to children's citizenship, participation, and equity have often been raised, especially at the end of the millennium in early childhood education and care (ECEC) in Finland (see Karila 2012). Children's citizenship is conceptually and practically multifaceted because its implementation tends to violate assumptions connected to the generational relationship. According to the United Nations' Convention on the Rights of the Child (UNCRC; UNICEF 1989), children's participation has been defined as children's initiatives and opportunities for influence in matters that concern their lives and everyday communities (Horgan et al. 2017; Lundy 2007; Shier 2001). Yet, the participation of young children is linked to views of the power hierarchy between children and adults, which situates children as "not-yet-citizens", making them subject to protection, and justifies the use of power and decision making by adults in matters concerning them (Lansdown 2010; Lister 2007; Vranješević 2020).

Therefore, essential questions related to young children's participation center around dilemmas of the adult–child hierarchy, the rights of the individual versus those of the group, and the rights of the majority versus those of the minority (Theobald 2019). ECEC is a significant everyday community for young children in Finland. Therefore, such institutions need to pay attention to ways of tackling the vulnerabilities that diverse children face in their everyday surroundings within which they negotiate their participation and belonging (Jensen and Iannone 2018; Vandenbroeck 2010). For this reason, we interviewed students who participated in education on children's rights and participation. In this study, we examine what kinds of counter discourses can be differentiated in these interviews about

the realization of children's participation. Our study data include individual interviews and group discussions collected with 31 students.

## 2. Children's Participation and Lived Citizenship in ECEC

Children's participation rights stem from the UNCRC, in which their rights can be roughly categorized under three Ps: provision, protection, and participation (see Hammarberg 1990). Participation is sometimes considered less important than the rights of provision and protection, but we agree with Alderson (2010) that without participation, children's rights do not materialize even in protection and provision. That is why listening to children's views and opinions and enhancing their involvement in issues that concern them, as Articles 12 and 13 of the UNCRC suggest, are crucial. Children's participation is an established moral value in our society, but at the same time, it has become part of society's effort to control children and "to include them in strivings for efficiency, flexibility and quality improvement" (Strandell 2010, p. 180). Thus, its realization is a tense and complex question (Theobald 2019).

Young children are often considered to be future rather than current citizens. Because of their age, it is believed that they are unable to properly form opinions, although studies have shown how the participation of young and speechless children can be supported in an inclusive democracy (see Donaldson and Kylmica 2016; Vranješević 2020). Overall, researchers have drawn attention to the fragmented conception of "the child" in the UNCRC. First, the child is emphasized as the holder of rights; second, the child is the object of protection; and third, the child is an active participant (Hakalehto 2016; Vranješević 2020). These discrepancies seem to place children in the role of an object of protection who has the possibility to have control, but this controversy means that children's participation rights can be limited (Hakalehto 2016; Vranješević 2020; Warming 2019). However, children's right to protection cannot be realized if they are not consulted or they cannot influence how their rights related to protection or wellbeing are implemented (Alderson 2010; see also Vranješević 2020). Several researchers have criticized views that underestimate children's abilities and emphasize their irrationality and dependence because these conceptions invalidate children's participation rights and citizenship (e.g., Alderson 2010; Hakalehto 2016; Larkins 2014; Lister 2007).

As in Finland, Nordic ECEC has been characterized as being built on two basic pillars, namely the Nordic welfare state model and child-centeredness (Einarsdottir et al. 2015). The Nordic welfare model emphasizes the equality of every citizen and the prohibition against discrimination (see Non-Discrimination Act 2014 for the legislation in Finland). In the ECEC in Finland, this is realized by guaranteeing universal services to everyone, i.e., the Act on Early Childhood Education and Care (2018) stipulates a subjective, equal right to ECEC for every child (also see Karila 2012). Child-centeredness, on the other hand, emphasizes warm relationships, equity, solidarity, democracy, equality, freedom, and emancipation (Einarsdottir et al. 2015). In the Finnish Core Curriculum for ECEC (EDUFI 2022, pp. 21–22), child-centeredness is strongly linked to the child's right to be heard, seen, considered, and understood as an individual and a member of the community. The participatory rights originating from the UNCRC are strongly present in the Finnish Core Curriculum for ECEC, which states that "Personnel attends that every child has a possibility to participate and influence" (EDUFI 2022, p. 29). All in all, ECEC educational institutions can be regarded as having an ethical responsibility to promote inclusion, democracy, and equal citizenship (Åmot and Ytterhus 2014).

Yet, setting participation and democracy as values in ECEC creates tensions in the implementation of the curriculum (Einarsdottir et al. 2015; Theobald 2019). One dilemma concerns children's position as either being or becoming (e.g., Karila 2012; Lister 2007; Warming 2012):, namely, this concerns the goal to raise children to become future democratic citizens or to implement democratic values in the children's current everyday lives? This dilemma is related to "adultism", the generational order in which children are viewed as vulnerable and incompetent and adults as autonomous, independent, and competent actors

(Vranješević 2020; Wall 2022; Warming 2012, 2019). Consequently, children are thought of as "citizens-in-waiting" or "citizens-in-the-making", not as citizens in the here and now (Lister 2007; Warming 2012, 2019). Children's participation is limited by their subordinate position compared to that of adults. In practice, this can be visible in the conventions related to children's participation that remain ostensible. For instance, listening to children can be a hidden exercise of power, whilst actually children have to adapt to adults' decisions and opinions (Millei 2012; Moran-Ellis and Sünker 2018; Raby 2014).

Children's participation has been framed as part of citizenship education, in which participation is determined by adults and institutions and is often representative (Warming 2019). In these representative groups formed to increase children's participation and ensure that their voices are heard, children's opportunities for real influence are generally limited and their initiatives are without impact (Kiili 2016; Warming 2019). Biesta (2011) suggests that instead of citizenship education, we should talk about learning democracy. To understand what democracy is, children must feel that they receive democratic treatment in their everyday environments (Biesta 2011; Lister 2007; Warming 2019). Lansdown (2010) argues that adults often underestimate children's ability to participate in decision making, although most often the issue is that children are not provided with sufficient opportunities to practice decision making and taking responsibility. According to her, adults do not recognize or promote children's opportunities to be involved, failing to renounce their own control for the sake of the realization of participation.

There are more conceptual opportunities to consider when framing and defining children's participation. The concept of "lived citizenship" refers to how people in their everyday environments understand and negotiate their rights, responsibilities, belonging, and participation (Fichtner and Trần 2020; Larkins 2014; Lister 2007; Warming 2019). Additionally, the difference-centered theorizing of children's citizenship has called for a citizenship debate instead of despising children's citizenship (Larkins 2014; Lister 2007; Warming 2012). The reconstruction of social relationships between children and adults difference-responsively has also been argued about in the discussion concerning "childism" (Biswas et al. 2023; Wall 2022). Instead of seeing citizenship as the (adult) capacity for autonomous decisions, difference-centered approaches emphasize (children's) agency as relational, i.e., in relation to others (Moosa-Mitha 2005; Warming 2012).

Indeed, in a difference-centered relational approach, children's citizenship is seen as built on everyday practices and relationships with other people and communities (Larkins 2014). When citizenship is understood as relational, it is described as children's membership in and a sense of belonging to communities important to them, such as ECEC and school, in which children's voices and perspectives are seen as legitimate and valuable for participation (Lister 2007; Moosa-Mitha 2005; Moran-Ellis and Sünker 2018; Warming 2012, 2019). Warming (2019, p. 336) emphasizes that children's participation should not be merely defined as autonomous decision making, but it "is about being included in social practices of community [. . .] and a subjective feeling of belonging".

There is also a tension between valuing children's individual agency and valuing democratic goals in the community (Biesta 2011; Einarsdottir et al. 2015; Kampmann 2004; Karila 2012; Theobald 2019; Zeiher 2009). This is related to the question of the individualization of childhood compared to the institutionalization of childhood (Kampmann 2004; Zeiher 2009). Institutionalization is linked to the goal of guaranteeing educational opportunities for everyone regardless of gender, social class, or ethnicity. However, it also gives institutions the power to define goals from the perspective of social and economic development. Zeiher (2009) estimates that, along with individualization, institutionalization nevertheless supports equity, democratic goals, and equal learning opportunities.

Children are easily presented as a unified group (Lister 2007); however, they are not. Nor is childhood the same for every child. Intersectional thinking reminds us that each of us is simultaneously defined by several identity categories, which can produce inequality and discrimination in different ways (Konstantoni 2013; Lister 2007). Inequality can be found not only in age but also in other differences among children, such as gender, culture,

religion, family structure, language and linguistic ability, or emphasis on academic skills (Horgan et al. 2017; Konstantoni 2013). For example, a young child can simultaneously be evaluated based on their age, gender, ethnicity, and abilities.

Individual or representative participation can be seen to be in conflict with issues of care, responsibility, and the common good (Einarsdottir et al. 2015; Konstantoni 2013). Representative participation, such as student unions, does not necessarily promote mutual cooperation among children in the best possible way (Kiili 2016), since conflicts can arise between communal and individual participation, as well as between majority and minority rights (Theobald 2019). In ECEC, as in the field of education more broadly, deficiencies have been found in the practices of inclusion, equality, and the prevention of marginalization (Arvola et al. 2017). Even small children can participate in discriminatory practices, and they observe and notice inequality and discriminatory practices around them, so educators should consciously promote equality (Konstantoni 2013). Accordingly, the emergent challenge for educational institutions is to promote participation and to reduce and eliminate discrimination by identifying and responding to the increasingly diverse needs of children and young people (Arvola et al. 2017; Jensen and Iannone 2018).

In this study, we examine the extent to which the dilemmas of children's participation, which have been presented in previous studies, are pertinent in present and future educators' speech. The earlier research portrayed here comprises mainly theoretical considerations (e.g., Lister 2007; Moosa-Mitha 2005; Moran-Ellis and Sünker 2018; Wall 2022; Warming 2012) or is conducted from the perspective of children and based on ethnography or participatory methods (e.g., Arvola et al. 2017; Fichtner and Trần 2020; Horgan et al. 2017; Konstantoni 2013; Larkins 2014; Åmot and Ytterhus 2014). However, how present and future professionals consider children's participation in education is scarcely examined. This study aims to fill in this gap in the research. The aim of this study was to increase our understanding of and knowledge about enhancing children's rights and participation in pedagogical settings in ECEC while acknowledging the dilemmas of children's participation. To identify the dilemmas, we approach the interview data of this study through the concept of counter discourse. The research question is as follows: What kinds of counter discourses about the realization of children's participation can be identified in the interviews with future and present education professionals?

## 3. Materials and Methods

### 3.1. Data Collection

Data collection for the study was undertaken during the teacher education development project *OIVA—Children's rights and participation in ECEC, preschool and first grades*, which was carried out from 2018 to 2021 at the University of Jyväskylä. The project developed a study module (25 ECTS) focusing on children's rights and participation that was offered to degree students in educational sciences and professionals in the field of ECEC and pre-primary and primary education. The study module provided the participants with the opportunity to develop and renew their skills and knowledge of children's rights and participation in accordance with the principles of the Finnish Core Curriculum for ECEC (EDUFI 2022) and the Finnish Core Curriculum for Basic Education (EDUFI 2014). For degree students, the study module was voluntary.

In total, 56 students participated in the study module. They were all women whose ages ranged from 20 to 60 years (mean age of 36 years). Among the students, 27 (48%) were degree students and 29 (52%) were employees. Most professionals worked in ECEC units, but there were also professionals from other educational institutions. Some of the degree students also worked in an educational profession. The educational background of the students varied from the secondary to the doctoral level.

To evaluate the study module, we conducted individual interviews at the beginning and end of the study module, and group discussions took place in the middle of their studies. The interviews and group discussions form the data for this study (see Table 1). In total, 31 students participated in the data collection. In all cases, the interviewer or the

discussion moderator was a person not working as an educator in the project. Participation in the study was voluntary, and the participants had the opportunity to withdraw from the study at any phase. All interviewees and group discussion participants gave their informed consent to the data collection for research purposes. The study followed the guidelines of the Finnish National Board on Research Integrity TENK (TENK—Finnish National Board of Research Integrity 2012), which were also communicated to the students who participated in the module, and they were given a research privacy notice.

**Table 1.** The datasets.

| Data Source (Abbreviation Used) | Number of Participants | Data Collection Method |
|---|---|---|
| Initial interview (II) | 10 | Individual interview |
| Final interview (FI) | 11 | Individual interview |
| Group discussion (GD) | 15 | 3 group discussions, each with 5 participants |

The initial interviews were conducted with 10 students, among whom five were degree students and five were professionals working in ECEC/education. The selection of the interviewees could be characterized as systematic sampling, since we invited every sixth participant from both student groups to take part in the interview. All of them agreed to be interviewed.

In the final interviews, we changed the sampling method. Since one of our interests was the change in the students' learning, we invited half the participants from the initial interview to take part in the final interview. The selection of these interviewees was based on drawing lots. As we also wanted to give voice to as many students as possible, we complemented the sample with students who had not participated in the initial interview or in the group discussions. We also chose these participants based on systematic sampling. Thus, 11 students were interviewed at the end of their studies; five of them had also taken part in the initial interview. In total, the interview data consist of 21 individual interviews with 16 students.

The initial interview focused on the starting points and learning needs of the study module participants and inquired about their work and study histories, motivation for applying for the study module, expectations related to their studies, and the prior knowledge and skills they had about children's rights and participation. The aim of the final interviews ($n = 11$) was to provide the participants with an opportunity to freely give feedback and reflect on the development of their competences. The interview questions dealt with the content and implementation of their studies and what benefits their studies could have for the current or future careers of the participants and their work community. The interviewees were also asked what issues they considered relevant to the development of their expertise.

In addition to the individual interviews, we organized three group discussions with five participants in each discussion in the middle of the study program. To ensure we gathered data from as many students as possible, we encouraged those students who had not participated in the initial interview to take part in the group discussions and thus, in the quality evaluation of the project. In total, 15 students volunteered. The guidelines were as follows: "Discuss now as a group and consider what you have learned about children's rights and participation during your studies in the OIVA study module?" The discussions, which lasted about 20 min, were video recorded.

Both the interviews and group discussion recordings were transcribed verbatim and pseudonymized. The interviews amounted to 144 pages. In the excerpts, which will be presented later, we use the terms initial interview (II) and final interview (FI), interview numbers 1–11, and the terms professional and degree student (for example, FI2, degree student). The group discussions resulted in 24 pages of text. In the excerpts, the discussions

are coded with the abbreviation group discussion (GD) and the numbers 1–3 referring to the group, and the speakers are identified by the numbers 1–5 (e.g., GD3, speaker 3).

### *3.2. Data Analysis*

In the analysis of the data, we applied discourse analysis following a social constructionist approach (e.g., Burr 2015; Nikander 2007) and used discourse as the analytical concept. Although the definition of discourse varies in research, it is commonly considered to be a medium of action situated in a specific context and constructed from a range of discursive tools (Potter 2012). Thus, discourses perform actions and are constructive. In other words, they build versions of different phenomena, such as people, actions, social organization, and psychological worlds, through different linguistic means, for example, words, vocabularies, grammatical structures, and categorizations (Potter 2012). Discourses are also constructed in specific institutional and interactional contexts (Potter 2012), such as the higher education and ECEC contexts, as well as the research interview in this study. In each context, discourses can be understood as reflecting the cultural resources available to people to make sense of the world and phenomena in it (Burr 2015; Wood and Kroger 2000). Discourse analysis findings are, therefore, not considered to reflect the internal structures of individuals' minds, attitudes, or the like, but to produce knowledge about their culture (e.g., Wood and Kroger 2000).

The analysis started with a close reading of the interview and group discussion data to recognize and differentiate repeated patterns related to children's participation. Soon, it became evident that the interviewees and group discussion participants commonly referred to and acknowledged the norm and requirement of enabling children's participation in ECEC and school. However, a certain dualism and tensions characterized their talk about children's participation. In other words, the speakers typically described issues that counteracted the actualization of the norm or presented it as contingent. We decided to focus the analysis on these descriptions of tensions and attended to the linguistic features, such as modal terms and subject positions, in these descriptions (see, e.g., Wood and Kroger 2000). Through joint discussions among the authors, we constructed three counter discourses based on the identification of tensions in children's participation and the linguistic features and vocabularies of the descriptions of the tensions.

## 4. Results

Based on the analysis of linguistic features, we identified three different counter discourses from the informants' talk on children's participation that ran counter to the idealized discourse and constructed participation as (1) unrealized; (2) adult-defined; and (3) elusive. In the three counter discourses, the talk about and the challenges and successes of the actualization of participation varied and children were positioned differently.

### *4.1. Unrealized Participation*

The discourse about unrealized participation was differentiated from talk concerning children's participation rights as an obligatory or juridical requirement and norm, but at the same time, participation was considered not to be the reality in ECEC, school, or society. Children's participation rights were discussed in general terms, and children were talked about as a generic, homogeneous group through a variety of words, including "all children", "the child", and "every child". Participation was linked with terms such as "equality", "respect", and "listening to", as well as described as "an idealistic thing" or "idealism" that is not truly reflected in educational organizations or society.

In the following excerpt, the non-realization of children's participation is demonstrated both in the ECEC professional's reflections on the gap between the organizational practices of ECEC and the legislative requirement of children's participation, as well as in her consideration of what needs to be done.

### 4.1.1. Excerpt 1

The child's participation is actually written in law and [guiding] documents. How much can it be seen in all [that is done], how we formulate the forms [related to pedagogy] or plan those [pedagogical] processes? Indeed, the child is not heard anywhere in them. It was a revelation [in the study module] that this is actually the value base, [...] this is where we start from. But is it [children's participation] really visible? [...] We need to start from the fact that the child is not anymore subordinate. The child's equality, listening to [the child], how can the child influence everyday life, for example, in ECEC or at school?

(FI9, Professional)

In excerpt 1 and more generally in this discourse, the ECEC professional quoted above mentions the norm of children's participation, which is grounded in ECEC legislation and the UNCRC, but at the same time, the speaker believes the norm is not yet reflected in the reality of early education. For example, the professional points to the missing voices of children in different documents related to pedagogical planning in ECEC, although the legislation requires that children are heard. Moreover, using a rhetorical question ("How much can it be seen...?") as a discursive tool, the professional underlines the gap between the legislative requirements and ECEC practices and thus, the non-realization of children's participation. Furthermore, her suggestion about considering children not as subordinates but as equals is presented as something for which we should strive (the modal expression "we need to start"), not something that is a reality. Similarly to the professional in excerpt 1, many interviewees considered the study module to be "eye-opening" or enlightening regarding their understanding of the normative role of children's participation.

The next excerpt also illuminates how children's participation is talked about as something that should or perhaps could be real in the future but not a reality at present. Thus, participation is constructed as an unrealized ideal.

### 4.1.2. Excerpt 2

That was very crucial, if children can experience participation in their daily life maybe in the present and future ECEC culture [...] so there'd be a new generation for whom participation would be natural. This should be seen as a broader issue [...] And indeed, I thought that it [achieving such participation] is such an idealistic thing that it should be somehow mobilized in other ways too. [...] It should not be connected only to ECEC or school but to wider society too.

(GD3, Speaker 4)

Through various modal terms and expressions ("if children can", "would", "should") and by referring to the "new generation", the speaker implies that children's participation is not a reality or "natural" in the present but as something that needs to be worked on in educational institutions and in society at large to become a reality in the future. The speaker also underlines the gap between the reality and "natural" participation and the distance to the goal of such participation by denoting the participation as an "idealistic thing".

Thus, while the discourse of unrealized participation acknowledges children's participation as normative, it presents participation as an ideal that does not yet exist in educational institutions or society at large. The discourse contrasts the current opportunities for children to participate and the legislative requirements concerning participation. Achieving ideal or true participation is seen as demanding attention and work not only in education but also more broadly in society.

### 4.2. Adult-Defined Participation

In the counter discourse of adult-defined participation, the participants discussed how, in actual pedagogical practices, adults impose or place limitations on children's participation. Thus, in the discourse, adults are positioned as having the possibility or

responsibility to control children's participation. In the next excerpt, one of the speakers in the group discussion highlights how, in daily ECEC practices, the adult defines children's participation.

### 4.2.1. Excerpt 3

But they are thus allowed to choose only that one thing; the adult has decided beforehand that this is the thing this week that children can have a say about. So, is it genuine participation when the adult has already defined what it is?

(GD2, Speaker 2)

The excerpt starts with "but", which demonstrates that what follows contrasts with what has been previously said (the ideal description of participation) (see Tannen 1993). The speaker underlines children's limited possibilities to influence ECEC practices, for example, by using the adverb "only" and the numerical term "one" and thus, their subordination to the adults in ECEC. As in the excerpt, the discourse on adult-defined participation presented children's participation as strongly controlled by adults: adults were positioned as subjects in defining when, how, and in which matters children were allowed to express their views. In the interviews, the adults were presented as "definers" and "limiters" of children's participation, and this was contrasted with "genuine" participation. The counter discourse also highlights the tokenish nature of children's participation: children are listened to but they do not have any real influence.

However, the limitations on participation were also considered essential and thus, justified for the sake of children's futures.

### 4.2.2. Excerpt 4

When we talked about participation in the beginning, I thought that [...] they [the children] can do anything and everything that is possible. But safety is also an essential part of it [participation]. Adults need to limit it a bit and make it possible for children to grow up to become members of society. It's not such that they should be allowed to do everything and all should be possible. But there is also the perspective of safety and in this sense [...] their future will be guaranteed.

(FI8, Degree Student)

The speaker describes her first impression of children's participation as limitless self-determination. What follows is the denial of this view or challenging it with the sentences starting with "but". The necessity of limiting what children are allowed to do is also produced using the modal term "need". Thus, the discourse constructs children as "becoming", as future citizens. Adults are positioned as responsible for limiting and thus protecting children to ensure their safety, best interests, and future wellbeing.

All in all, at the center of the counter discourse of adult-defined participation, there is the question of hierarchy and generational order and thus, the structural power between adults and children in educational institutions. Here, children's participation is seen as limited by adults because of the prevalent and unquestionable adult–child hierarchy that bestows on adults permission to limit children's participation, especially when this is done for the sake of children's safety and futures.

### 4.3. Elusive Participation

The counter discourse of elusive participation constructs children's participation as elusive for adults or in some way unreachable. The discourse was differentiated in that it first constructed adult knowledge and perspectives as essentially different from the child's world and second characterized actualizing the participation of specific children or child groups as beyond the realm of possibility for adults. The next excerpt illustrates the former type of talk.

### 4.3.1. Excerpt 5

Interviewer: What might be the one, singular, and most important thing that you have learned during this education?

Interviewee: [. . .] it's how we [the students in the program] [. . .] have stopped and pondered in small groups together what the child is in the end, how we reach the child's thoughts. And what is in the best interest of the child and what is good for the child and the rights of the child? [. . .] So, we know in terms of the big picture how the child develops, and as the people in the field, we know what the child needs. But I think that there is humbleness in the fact that each child carries her own world with her, she has [. . .] her own matters. They are not such that she can list them with bullets, that these are then Tuula's [the participant uses her own name], my important things.

(FI9, Professional)

Excerpt 5 depicts the utmost limit between the child and adult by linking adult knowledge to the "big picture", and thus, the general idea of child development and needs, while pointing to the child's knowledge as personal, individual, and internalized in such a way that the child is not able to communicate it to the adult. So, the adult is considered to be unable to understand what an individual child's needs or wants are and to actualize their "rights", "best interest", and "participation" accordingly. The speaker thus draws a line between adult and child knowledge before which the adult needs to be "humble".

The next two excerpts demonstrate how, in the discourse of elusive participation, ideas of representative and equal participation among all children were challenged by considering participation to be elusive and unattainable, especially for certain groups of children.

### 4.3.2. Excerpt 6

The situation of migrant children in Finland is challenging at the moment. [. . .] There were children who didn't understand the Finnish language. So, it was rather difficult for them to have an equal position in anything because they couldn't participate in their ECE center. If there was a children's meeting where the name of the group or something else was discussed, [. . .] when they couldn't speak any single Finnish word, so they couldn't participate at all in that.

(II8, Professional)

### 4.3.3. Excerpt 7

Then, even the youngest children, they too have a right to their own, their own voice and participation, not only those who are able to produce their views verbally and narrate them. It's a challenge, but kind of a good challenge that you want to take on. It's a lot different when it comes to older children, but the most important thing [for the youngest children] though, is that's where it all starts.

(FI2, Degree student)

In this counter discourse, the particular groups of children for whom participation is a challenge are openly named: for instance, the refugee and migrant children face difficulties related to participation in daily practices and activities in ECEC in excerpt 6. In turn, the speaker in excerpt 7 evaluates the youngest children's positions in ECEC and how their participation remains easily unnoticed. Linguistic features and vocabularies in the discourse, such as "is challenging" (in excerpt 6) and "difficult for them to have an equal position" (in excerpt 6) refer openly to challenges and inequality. In excerpt 6, the difficulties and inequality are emphasized by the repeated use of negation four times and modality "could" with negation three times. In excerpt 7, the speaker emphasizes young children's "right" to participation and "their own voice" and contrasts young children's

participation with that of "older children" who are able to express their views verbally by setting young children's participation as "the most important".

Consequently, the discourse highlights in detail what kinds of challenges might face the participation of children in the margins or vulnerable groups, for example, the youngest ones, immigrant children, and children with special needs. The discourse frames children's participation primarily as based on verbal communication. By considering children as unable to communicate their inner world to adults or specific child groups as incapable of doing this, the discourse challenged the ideal of children's equal participation or participation in general.

## 5. Discussion

In this study, we investigated the counter discourses of future and present education professionals applied to challenge the ideal and norm of children's participation rights in everyday ECEC and school environments. We identified three counter discourses: unrealized, adult-defined, and elusive participation that questioned the ideal in different ways. All the counter discourses, however, shared the same function (see Wood and Kroger 2000): they constructed children's participatory rights as an ideal norm that is not actualized in practice in ECEC in Finland.

The first counter discourse of unrealized participation highlighted how legislation and guiding documents, which create a legal and professional framework for promoting child participation, remain just written statements without reality or meaning in the daily practices in ECEC or society. Children's lived participation and citizenship were depicted as unattainable or only to be actualized in the future. Moreover, the discourse pointed out the need for society in general to change to ensure children's "true" participation. Thus, the responsibility to make a change was (partly) removed from the ECEC and placed on the shoulders of society.

From the perspective of children's participation rights, what matters is how the statutory nature and value of participation is translated into action (Theobald 2019; Warming 2019). Children's participation can be a clear aim in ECEC, as illustrated in the Finnish National Core Curriculum for ECEC (EDUFI 2022), but its actualization may be challenging because of the vague definition of participation and the lack of more specific pedagogy for participatory practices (Niemi et al. 2016). In the counter discourse of unrealized participation, this dilemma is emphasized.

The adult-defined participation counter discourse frames children's participation as limited by the adult–child power asymmetry and generational order in ECEC in Finland. The discourse includes an interpretation of children as future citizens justifying adults' decision making concerning children. The adult has the power to define when, how, to what extent, and in what matters children can and are allowed to participate (Millei 2012; Moran-Ellis and Sünker 2018; Raby 2014; Warming 2019). A key dilemma, which the counter discourse displays, is to what extent children are seen as targets of adult control, protection, and care or recognized as valued members of and influencing their communities (Lansdown 2010; Theobald 2019; Warming 2019). Furthermore, in a certain way, the whole ideal of children's participation and children's citizenship becomes defined erroneously as unlimited self-actualization and thus, questionable in the discourse.

The counter discourse of elusive participation was characterized by the contradiction between, on the one hand, adult and child worlds and knowledge and, on the other hand, the equity of participation versus inequality and polarization. Unequal participation was seen as a shortcoming of professionals because they did not have access to the child's world and knowledge. In this sense, paradoxically, listening to children's views could be considered as elusive and unreachable from the adult perspective. Second, adults relying on verbal communication and not identifying this as a mechanism producing inequality nor having the means to realize children's participation in heterogeneous child groups in ECEC was constructed as a reason prohibiting the equal actualization of children's participation. Presently, in the Finnish ECEC, increased global migration has led to greater diversity

among children, while the scarcity of financial resources and heightened racist voices have enhanced polarization (Arvola et al. 2017; Jensen and Iannone 2018). Several researchers consider the equality of participation of diverse vulnerable groups of children to be an important issue (e.g., Konstantoni 2013; Lansdown 2010; Lundy 2007; Moran-Ellis and Sünker 2018; Vandenbroeck 2010; Warming 2019). Furthermore, the vulnerable groups of children are emphasized separately in the UNCRC, because they are in greater danger of discrimination and other abuses related to their rights (Hakalehto 2016; UNICEF 1989).

The counter discourses can be interpreted as constructing different obstacles to children's participation. At the same time, different counter discourses show the cultural and social interpretation repertoires of young children's participation in Finnish ECEC and culture more broadly. Hence, the counter discourses found in the study raise fundamental ethical questions about the actualization of children's participation. First, adult positions and power in the counter discourses (cf. Moran-Ellis and Sünker 2018; Raby 2014) are reminders of the adultism of our present-day societies. Therefore, the concept of lived citizenship and the theorization of the difference-centered approach are crucial when considering children's citizenship and participatory rights because citizenship and democracy are negotiated and restricted by everyday practices (Fichtner and Trần 2020; Larkins 2014; Lister 2007; Warming 2012, 2019). Moreover, participation is linked to questions of exclusion, discrimination, and equity between different groups of children (Konstantoni 2013; Warming 2019). Theobald (2019) points out the controversy of the democratic perspective: whether the rights of the majority exceed the realization of the rights of minorities.

## 6. Limitations

When evaluating the results of the study, one should consider the limitations related to its implementation. In terms of discourse analysis, our study focused on the analysis of linguistic features of the participants' speech, variation, and language structure, (e.g., Wood and Kroger 2000). The results revealed counter discourses related to children's participation that might function as frames for (early childhood) education students and professionals more generally in Finland to approach children's participation in ECEC practices. However, the study participants could have been a priori interested in children's rights and participation, since they had also been interested in attending the study module. The counter discourses could have been even more comprehensive if data had also been collected from individuals who did not attend the study module. That said, the selectivity of the participants may have also brought richness and depth to the material, since the participants had had several opportunities to consider and learn about children's participation. Many of the participants reflected on the change during the study module from understanding participation as a narrow and specious mapping of children's wishes to a holistic view of the importance of listening to and respecting children's views. The counter discourses also resonate with earlier studies on dilemmas of children's participation (Karila 2012; Lundy 2007; Theobald 2019; Warming 2019).

Moreover, the gender and ethnic background of the research participants can be seen as a limitation of the study, as all participants were women and of native Finnish background. So, the study did not reach the views of men working and studying in education or the perspectives of professionals representing ethnic minorities. The underrepresentation of these groups is not uncommon because the professionals in education in Finland, especially in ECEC, are mainly women (Alila et al. 2014). Typically, they are also of Finnish background. Thus, it is noteworthy that the research participants considered children's participation from their own cultural perspective.

A further limitation is using the diverse data sources for the same discourse-focused analysis process, in which case the specific nature of the different data sources remained partly underutilized. However, the consistency of the occurrence of the identified counter discourses in the data collected through different methods and at different phases of the study strengthens the credibility of the analysis.

## 7. Conclusions

The counter discourses identified in this study suggest that the activities of professionals and other workers in Finnish ECEC communities are partly built on conflicting assumptions about what is meant by children's participation (cf. Theobald 2019). Nonetheless, promoting participation has a "hard" legal basis that can be found in the UNCRC, as well as the Act on Early Childhood Education and Care (2018), the Finnish National Core Curriculum for ECEC (EDUFI 2022), and the Non-Discrimination Act (2014) in Finland. Promoting participation is not optional; the role of the ECEC is to enhance the participation of children. If children's initiatives and views are to be taken seriously, it is necessary to question adult dominance and apply a democratic and difference-centered perception of children and pedagogical skills that promotes the equal participation of diverse and vulnerable groups of children.

**Author Contributions:** Conceptualization, E.S. and M.M.; Formal analysis, E.S. and M.A.; Investigation, E.S., M.M. and M.A.; Methodology, M.A.; Writing—original draft, E.S. and M.A.; Writing—review and editing, M.M. All authors have read and agreed to the published version of the manuscript.

**Funding:** This study was funded by the Ministry of Education and Culture in Finland.

**Institutional Review Board Statement:** The study did not require the institutional review board statement according to the guidelines of the Human Sciences Ethics Board in the University of Jyväskylä.

**Informed Consent Statement:** All participants of the study were asked and gave their written informed consent.

**Data Availability Statement:** The datasets are not publicly available for ethical and privacy reasons.

**Acknowledgments:** We want to warmly thank the participated students of the OIVA study module and the members of the OIVA-team.

**Conflicts of Interest:** The authors declare no conflicts of interest.

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
