# Peer review of "Dilemmas Related to Young Children’s Participation and Rights: A Discourse Analysis Study of Present and Future Professionals Working with Children"

_socsci, doi:10.3390/socsci13010027_

Round 1

Reviewer 1 Report

Comments and Suggestions for Authors

The article is quite interesting as it offers an insight into counterdiscourses on children's participation. The theoretical and normative backgrounds are well covered. The methodology is sound and well presented. The results are clear and systematic. The literature is up to date, and the limitations and the conclusions are relevant. It was a pleasure to read this article, thank you.

  1. the question addressed by the article is about the kinds of counterdiscourses about children's participation that can be found in the responses of 31 students who are present and future professionals in the education system in Finland.
  2. the topic is relatively original as usually it is mainly children who are asked about their own participation, and therefore asking present and future professionals in the field of education fills a gap.
  3. This article adds a reconstruction of discourses along a social constructionist approach, which is well presented in the methodology.
  4. The methodology is systematic and relevant.
  5. The conclusions are consistent with the evidence and arguments presented and they address the question posed.
  6. The references are appropriate and up to date.
  7. Table 1 is clear and does not require changes.

Author Response

We wish to thank the reviewer for the positive feedback on our manuscript, showing close reading of the manuscript. All corrections are marked in text with red colour.

Reviewer 2 Report

Comments and Suggestions for Authors

interesting and important article that should be published following minor amendments as noted in attached 

Author Response

We wish to thank the reviewer for the comments and advice. We found these insightful, showing close reading of the manuscript. Here is a more detailed list of the corrections made along the comments:

!. Abstract says participants were practitioners yet interviews were with students? Needs explaining: students of what from where? Line 40 to 42 explains this in introduction but abstract needs to note

We specified the abstract as follows: "In this study, we examined what kinds of counter discourses about the realization of children’s participation could be differentiated in interviews with present and future education professionals who took part in a study programme focusing on knowledge and skills regarding young children’s rights and participation. The data, which consisted of individual and group interviews with 31 participants, were analyzed with discourse analysis. Three counter discourses were identified: unrealized, adult-defined, and elusive participation. The discourses illuminated various dilemmas in children’s participation."

2. It would be good to know from the outset what the research questions are:

We have added the research question to the second paragraph of the manuscript on p. 2: “For this reason, we have interviewed students who participated in education on children’s rights and participation. In this study, we examine what kinds of counter discourses can be differentiated in these interviews about the realization of children’s participation.“

The research question is added into abstract and it is also modified in more detail in the last sentence before the methods section on p. 8: “To identify the dilemmas, we approach the interview data of this study through the concept of counter discourse. The research question is as follows: What kinds of counter discourses can be identified in the interviews with future and present education professionals about the realization of children’s participation?”

3. how the literature review was undertaken (search terms, time boundaries and geographical reach) and then how the literature review identified a gap that the research questions are aiming to fulfil .Lines 168 to 175 : this is the only part of the introduction that explains why the literature review undertaken is relevant to the article. There needs to be a closer link between the aim of the literature review and the subsequent exposure of gaps that the research here will fill.

We also specified into the last paragraph before the methods section the gap in research as follows: “The earlier research portrayed here comprise mainly theoretical considerations (e.g., Lister 2007; Moosa-Mitha 2005; Moran-Ellis and Sünker 2018; Wall 2022; Warming 2012) or is conducted from the perspective of children and based on ethnography or participatory methods (e.g., Arvola et al. 2017; Fichtner and Trần 2020; Horgan et al. 2017; Konstantoni 2013; Larkins 2014; Åmot and Ytterhus 2014). However, how present and future professionals consider children’s participation in education is scarcely examined. This study aims to fill in this gap in the research.”

4. The method needs to explain what the interview questions were as related to the research questions

The discourse analysis focuses on how certain meanings are constructed in talk, thus, the research question relates to the interview questions in a more general level and seeks to identify the variation in the participants' talk about children's participation. 

5. Line 187: why all women? How might that a influence the study outcomes and how might that have impacted on outcomes from discourse analysis? Is the ethnicity of the students interviewed relevant or not? If not, why? Does the cultural and economic background of the student population interviewed impact on their understanding of rights, participation? Maybe just a short mention to recognise that students will come with their own bias and histories that will impact on their understanding is relevant?

We have added the text explaining the ‘biases’ of the data on the limitations section on pp. 27-28: “Moreover, the gender and ethnic background of the research participants can be seen as a limitation of the study, as all participants were women and of native Finnish background. So, the study did not reach the views of men working and studying in education or the perspectives of professionals representing ethnic minorities. The under-representation of these groups is not uncommon because the professionals in education in Finland, especially in ECEC, are mainly women (Alila et al. 2014). Typically they are also of Finnish background. Thus, it is noteworthy that the research participants considered children’s participation from their own cultural perspective.”

All the corrections made are marked in red in the manuscript file.

Reviewer 3 Report

Comments and Suggestions for Authors

Thanks you for being invited to read and comment your paper. It is an interesting and well-written paper. 

My only comment to consider is about the concept specious in the counter discourse no 2. I can see the relevans of this discourse even without this concept. However, I dont think the concept is justified a d motivated enough as it stands write now. 

Here are some typos I found,  to consider:

Line 124 The reconstruction of social relations between children and adults difference ... Consider changing to relationships

Line 168 In this article, we examined ... rather In this study, we examined                                                                                                

Line 197  discussion moderator was an person  ... was a person                                                                                       

Line 264 on one hand   - but there is no second hand pointed out.                                                                                                                               

Author Response

We wish to thank the reviewer for the comments and advice. We found these insightful, showing close reading of the manuscript. All corrections are marked in text with red colour. Here are more detailed responses to the comments made byt the reviewer:

1. My only comment to consider is about the concept specious in the counter discourse no 2. I can see the relevans of this discourse even without this concept. However, I dont think the concept is justified a d motivated enough as it stands write now. 

We have deleted the concept specious and now the discourse is named as Adult-defined participation. We acknowledge that the English term is not capturing the actual meaning we aimed at.

2. Here are some typos I found,  to consider:

Line 124 The reconstruction of social relations between children and adults difference ... Consider changing to relationships ->corrected

Line 168 In this article, we examined ... rather In this study, we examined -> corrected

Line 197  discussion moderator was an person  ... was a person -> corrected                                                                                   

Line 264 on one hand   - but there is no second hand pointed out.  -> removed       

All the corrections made are marked in red in the manuscript file.